# Barriers Affecting Women's Access to Urban Green Spaces during the COVID-19 Pandemic

Carolina Mayen Huerta * and Ariane Utomo

School of Geography, Earth & Atmospheric Sciences, University of Melbourne, Melbourne, VIC 3010, Australia; ariane.utomo@unimelb.edu.au
* Correspondence: cmayenhuerta@student.unimelb.edu.au

**Abstract:** During the COVID-19 pandemic, urban green spaces (UGS) have gained relevance as a resilience tool that can sustain or increase well-being and public health in cities. However, several cities in Latin America have seen a decrease in their UGS use rates during the health emergency, particularly among vulnerable groups such as women. Using Mexico City as a case study, this research examines the main barriers affecting women's access to UGS during the COVID-19 pandemic in Latin America. We applied a sequential mixed-methods approach in which the results of a survey distributed via social media in June 2020 to women aged 18 and older were used to develop semi-structured interviews with 12 women during October 2020. One year later, in November 2021, the continuity of the themes was evaluated through focus groups with the same group of women who participated in the interviews. Our results suggest that (1) prohibiting access to some UGS during the first months of the pandemic negatively impacted UGS access for women in marginalized neighborhoods; (2) for women, the concept of UGS quality and safety are intertwined, including the security level of the surrounding streets; and (3) women who live in socially cohesive neighborhoods indicated using UGS to a greater extent. Our findings highlight that while design interventions can affect women's willingness to use UGS by improving their perceived safety and comfort, they remain insufficient to fully achieve equity in access to UGS.

**Keywords:** geography of fear; gender; green spaces; violence; Latin America; quality; safety; mixed-methods; social equity; fear of crime

## 1. Introduction

A growing body of evidence suggests that exposure to urban green spaces (UGS) is critical to maintaining or increasing public health in cities, especially during emergencies such as the recent COVID-19 pandemic [1–7]. Access to UGS is associated with a multiplicity of benefits, including fostering social cohesion, improving air quality, enhancing the livability of neighborhoods, reducing overall stress and anxiety, and, more recently, lessening the feeling of isolation experienced during confinements [1,8,9]. Nevertheless, access to UGS is rarely neutral, predominantly in cities of the Global South [10–12]. Women, low-income individuals, people with disabilities, and ethnic minorities often encounter barriers, such as an absence of UGS in close proximity, fear of violence, poor quality or inadequate facilities, smaller available UGS, and high density, which during the pandemic was a critical hindrance to using these spaces [7,13–15]. The barriers mentioned above influence the possibility of visiting and enjoying UGS, which are essential to take full advantage of their restorative capabilities and improve people's health and overall well-being [3,16,17].

In Latin American cities, women are often underrepresented in public spaces, including green spaces [10,18]. Even so, research that focuses on the barriers preventing women from accessing UGS in the cities of this continent is scarce [19], particularly that which explores the restricted use of green spaces by women during the COVID-19 pandemic [7,11,15,20,21]. Unlike countries in the Global North such as the United States,

Norway, Italy, or England, where evidence on the use of UGS after the COVID-19 outbreak showed a significant increase for most segments of the population [22,23], the current data on women's use of UGS in countries such as Mexico, Peru, Brazil, and Argentina shows the opposite trend [20,21,24–27]. This is particularly concerning since access to UGS is of paramount importance for women, with far-reaching repercussions on women's mental health compared to men [17], as evidenced by Roe et al.'s [28] study on the implications of low exposure to UGS, which emphasizes the need to target this demographic. More broadly, beyond the context of the pandemic, cities aiming to become more inclusive should evaluate whether all population subgroups can access the existing green infrastructure, adapting the built environment to account for and promote the well-being of all residents [12,29–32].

Using Mexico City as a case study, this study aims to identify and understand the main barriers that have affected women's UGS access in Latin America and, therefore, UGS use during the COVID-19-induced crisis. Two central questions guide our paper: (1) What have been the main barriers to women's UGS access during the pandemic? (2) What factors could help increase women's use of UGS? These two questions are critical to understanding women's perspectives on UGS use and how to improve their experience while using these spaces. Furthermore, they have the potential to inform evidence-based policymaking designed to improve existing green areas and make access to these spaces more equitable in Latin American cities, which is of great relevance during a prolonged public health emergency such as the COVID-19 pandemic. To answer these two questions, we use a sequential mixed-methods approach, encompassing three stages in its data collection process and strategy of analysis, one quantitative and two qualitative [33,34]. Our research's qualitative components helped further explore the statistical results of the quantitative component, a technique widely used in exploratory studies [35,36]. First, we analyzed quantitative data from an online survey ($n$ = 1245) launched in June 2020. Second, through the analysis of survey results, we developed a qualitative instrument, semi-structured in-depth interviews with twelve women in October 2020. Finally, through the interviews, we were able to identify critical themes that required further exploration, which led us to conduct two focus groups with the same participants in November 2021.

### 1.1. The COVID Pandemic and UGS

In March 2020, the novel coronavirus outbreak was categorized as a global pandemic, prompting governments worldwide to take drastic measures to contain the spread of the disease and subjecting almost half of the world's population to stay-at-home orders [2,6]. Several cities began confinement periods during which mobility and contact between people were restricted [37]. Preliminary evidence on the effects of lockdown measures on the population has demonstrated that containment policies, such as stay-at-home orders and the closure of non-essential businesses and schools, had adverse effects on people's well-being by introducing psychological stress, anxiety, feelings of loneliness, anger, frustration, and in some cases, depression, especially for women [38–41]. In particular, as numerous reports have demonstrated, once SAR-CoV-2 was categorized as a pandemic, women experienced rising rates of domestic violence, loss of jobs, deterioration in mental health, and increased anxiety about paying for health services [42,43], which negatively impacted their fear of contracting the disease compared to men [44,45].

In the vast majority of affected cities, urban residents identified UGS as one of the only means to avoid a sedentary lifestyle, cope with the stress of the situation, and improve their physical health while maintaining distance from others, as most sports and recreational facilities closed during the first months of the emergency [14,46–50]. For example, during a study conducted in Oslo in March 2020, Venter et al. [22] showed that the use of green spaces had increased, becoming a measure of resilience to the pandemic that helped sustain the well-being of users. Pouso et al. [2] conducted an online survey in nine countries during the early months of the pandemic and found that people who accessed UGS reported more positive emotions and used these spaces to cope with lockdown measures. Furthermore, research in Belgium by da Schio et al. [23] revealed that respondents also placed more

importance on UGS after the pandemic began, showcasing the benefits perceived after their use.

The studies mentioned above support the claim that maintaining access to UGS and encouraging their use are essential measures for amplifying positive health behaviors in urban populations [51]. Even more so, considering that the growing body of evidence on the transmission of COVID-19 shows that outdoor spaces can be categorized as safe options for recreation and leisure [14,47,52]. However, despite the diversity of benefits UGS can provide, not all cities observed an increased UGS use during the health crisis [53]. For example, in Argentina, a survey administered in Buenos Aires (*n* = 1740) demonstrated that despite respondents associating UGS use with feelings of calmness and tranquility, they did not increase the frequency of use or time spent in UGS after the pandemic began [26]. In Mexico City, Google's COVID-19 Community Mobility Report revealed a decrease of nearly 34% in the use of parks during 2020 compared to 2019, with women significantly decreasing their use of UGS compared to men [24,25]. The latter is worrisome, as limited use of UGS during the COVID-19 pandemic has been associated with a deterioration in health, primarily mental health [54]. It is still unclear whether the fear of contracting or spreading COVID-19, the closure of UGS, restrictions in mobility, or additional factors—such as changes in the perception of UGS—could have potentially discouraged UGS visitation expressly for women.

### 1.2. Barriers Affecting Women's Access to UGS

The barriers that affect women's UGS access are varied and, in some cases, depend on the specific context of their location. For example, Manyani, Shackleton, and Cocks [18] have cited social norms in South Africa as obstacles that prevent women from using green spaces on ceremonial occasions. Another context-specific factor is extreme weather conditions, which can discourage the use of these spaces [55]. For instance, in Hyderabad, India, green spaces that lack adequate vegetation (i.e., tree cover) are not commonly used in the summer when the temperature exceeds 40 °C [56]. The racial or ethnic composition of neighborhoods can also affect the frequency of UGS use by minority women [57]. Comparable to these examples, there are several others in which barriers associated with specific social norms or geographies limit women's UGS access.

In perspective, aspects such as the aforementioned barriers are difficult to modify. Regardless, additional obstacles restrict women's access to UGS, such as distance, limited size, poor quality, perceived insecurity [13], and more recently, COVID-19 restrictions or fears [37]. Urban planners and policymakers' influence can modify these impediments to promote greater UGS use by women, thereby improving their health and well-being, particularly during a public health emergency [58–60].

#### 1.2.1. Distance and Size of UGS

The consensus in the literature has suggested that the distance to UGS is the main factor that enables or prevents UGS access because use is bounded by geography [16,61,62]. Distance plays a significant role in women's access to UGS, as evidenced by a study in two Chilean cities, which indicates that women walk shorter distances to access green spaces [63]. In particular, given that pandemic restrictions caused the closure of UGS or restrained residents' mobility in several cities, this factor was reported to have significantly influenced UGS use during COVID-19-induced lockdowns by reducing or eradicating nearby options and eliminating the possibility of visiting distant green spaces [4,43,46]. As an example, the results of a survey in La Palma and Zaragoza, Spain, where strict home confinement measures were put in place to prevent the spread of COVID-19, showed that people who live closer to green areas reported higher UGS use compared with those who live far away [64]. Furthermore, the results of another study based on a survey conducted in six countries suggested that, during the COVID-19 confinement period, people tended to visit green spaces at closer distances [37]. Given the extensive evidence on distance and its relationship to UGS access, particularly women's UGS access, it is safe to imply that, in

contexts with COVID-19 mobility restrictions, the lack of nearby UGS could be a critical barrier to women's UGS use [4].

In addition to distance, size is another factor behind the gendered patterns of UGS use. Studies carried out in densely populated cities, such as New York, revealed that users reported feeling calmer and accessing more often spaces that allowed social distancing, which is linked to the size of UGS [5]. Overall, larger spaces allowed residents to engage in activities while maintaining adequate distance from others, which was generally associated with a propensity to use them [20]. Nevertheless, pre-pandemic evidence shows that women tend to avoid larger spaces where they are not visible to others [19].

### 1.2.2. UGS Quality and Safety

For women, safety and quality—which are strongly correlated and often used to explain one another—are generally more significant barriers to UGS access than for men, resulting in higher UGS use for women who have access to UGS of higher perceived quality or who do not fear violence when using these spaces [12,65–69]. For instance, Ode Sang et al.'s [17] study in Gothenburg, Sweden, and a second study conducted in the 40 major cities in China by Carli [21] showed that gender has a strong effect on the perception and use of green spaces, with women placing more emphasis on UGS quality characteristics, such as cleanliness, maintenance, order, and beauty, than men. Similarly, survey results from the Spanish city of Carmona have indicated that women attribute a higher value to quality and security features in green spaces than men, leading to increased use by women in safer and higher quality UGS [70]. Furthermore, a study conducted in South Africa found that women felt more discouraged than men to use green spaces because of the lack of maintenance, the arrangement of vegetation, and safety concerns [18]. Knapp et al. [71], who explored the relationship between UGS use and quality in 31 green spaces within low-income Black neighborhoods in New Orleans, United States, found that signs of concerns, which are commonly associated with safety-related feelings, and attractiveness were significant predictors of use for female users, while these same variables were insignificant among men. These results are similar to those of a study conducted by Williams et al. [68], who noted that women of color are exceedingly affected by the perceived lack of quality in UGS; in this study, safety (measured by crime rates) was the sole indicator used to define UGS quality.

Valentine [72] has defined the "geography of fear" as how feelings of vulnerability affect women's choices, mobility, and use of public spaces. To that end, perceptions or feelings are more frequently used to measure UGS quality and safety than objective measures [73]. Specifically, preconceived notions of a neighborhood or the particular experiences of women in a space may have a more significant effect on UGS use than objective crime rates [74]. In Latin America, such geographies of fear result from structural inequalities [75], which are especially alarming at a time when collective anxiety has risen as a result of the COVID-19 pandemic, and when measures to counteract the negative impacts of the pandemic are desperately needed [9,32,43,76,77].

## 2. Materials and Methods

### 2.1. Phase 1: Quantitative Analysis

In June 2020, we conducted an online survey to assess people's use and perception of UGS during the start of the COVID-19 pandemic. The survey was launched through social media platforms, promoting it only to adults (18 years and older). The survey was completely anonymous and consisted of four sections containing multiple-choice and open-ended questions related to (1) sociodemographic characteristics, (2) UGS use and frequency of visits before and after the COVID-19 pandemic began, (3) rating of neighborhood and UGS characteristics' importance, and (4) health-related questions. For the current paper, we use a subset of the survey sample consisting of only respondents who identify as women. Of the four sections in the questionnaire, we used the information corresponding to the first three to carry out the subsequent analysis.

A total of 1914 women started the survey, and 1245 completed it. We received responses from all municipalities and income groups. All the questionnaires that exhibited 100 percent progress were considered, including those questionnaires in which the participants decided for some reason not to answer one question but continued to answer the rest. This decision was made following the reasoning behind Lopez, Kennedy, and McPhearson's [5] analysis of park use in NYC during the early stages of the COVID-19 pandemic. The authors decided to evaluate the responses of those who had completed most of the survey due to the added value of their answers for the study's findings. To achieve a higher level of participation, we avoided asking for specific information. For example, asking about the respondent's income was ruled out. Instead, we requested respondents to indicate their income brackets. This approach is standard for data collection over the internet, where people may be more reluctant to provide information [40].

First, descriptive statistics were used to obtain information on the sociodemographic characteristics of the women participating in the study's first phase. Second, given that the evidence indicates that the quality of UGS is a determining barrier to women's access to UGS [19], to understand whether the quality of UGS has influenced access during the pandemic, we ran three logistic regression models to explore the association between UGS access and UGS quality. UGS use was adopted as a proxy for UGS access based on the study by Van Herzele and Wiedemann [13] that identifies access as a precondition for the use of UGS. The first model included only the association between UGS use and respondents' perception of UGS quality in their neighborhood (good quality = 1, not good quality = 0). The second model accounted for potential confounders to calculate adjusted odds ratios, exploring if they were significantly associated with UGS use. Finally, the third model incorporated neighborhood characteristics, including the respondents' opinion of whether there are enough UGS in their neighborhood and whether their neighborhood is considered quiet.

Finally, we carried out a logit regression model to address the second research question and examine which factors are associated with UGS quality and could help increase women's use of UGS if present. The model assessed the effect of ten binary variables (cleanliness and maintenance, good lighting, walls, low noise, markets, toilets, ample size, events, police presence, and playgrounds) on the likelihood of respondents perceiving UGS in their neighborhood as them being of good quality or not. The significant variables were later explored in the qualitative components of the study. The predictor variables were tested a priori to validate that there was no violation of the assumption of linearity of the logit.

### 2.2. Phases 2 and 3: Qualitative Analysis

The findings of the first stage of the analysis were used to develop the second phase of the study, which consisted of semi-structured interviews that delved into women's barriers and enablers related to UGS access. The introduction of qualitative components makes it possible to recognize and articulate cultural contexts into policymaking, achieving higher levels of effectiveness in the analysis [78]. The rationale for including two sequential qualitative components in the study (QUAN → QUAL → QUAL) was to deepen the reasons behind the survey's key findings, enriching the robustness of the study's insights by ensuring that the follow-up qualitative data provided a better understanding of the survey results [79]. By integrating qualitative data, we were able to provide a depth and breadth that the quantitative approach lacked by itself [80].

We invited survey respondents to participate in in-depth interviews that would take place in October 2020. Twelve women (ages 20 to 59) were selected to participate in the interviews, which lasted between 45 and 80 min via telephone. Building on the work of Sargeant [81], who defines how to ensure the quality of participants in qualitative studies, we selected participants who could best inform our research questions and enhance our understanding of the barriers that affected women's access to UGS during the pandemic. Aligned with grounded theory, we used a maximum variation sampling strategy, selecting women from different social strata living in different city municipalities; some lived alone

while others with their families, partners, or friends [82]. Additionally, we asked interested parties if they had young children or lived with older adults to assess whether attitudes changed depending on their care responsibilities. The selection of participants intended to provide multiple perspectives since the selected women came from different backgrounds and had diverse living conditions.

Comparable studies, such as Noël et al. [48] in Belgium and McCormack et al. [83] in Canada, have also used in-depth interviews to better understand the association between UGS use and other variables. The type of analysis used in this stage was thematic, designed to find, review, and name common themes regarding the factors that encourage or discourage women's UGS use. Thematic analyses have been identified as a useful tool to investigate the patterns and commonalities among group participants [83].

Finally, in November 2021, this same group of women was divided and invited to participate in two 90-min focus group sessions via Zoom. The objective of the third stage of the data collection and analysis was to investigate the continuity of the previously identified themes once the restrictions associated with COVID-19 eased. Additionally, we inquired about characteristics related to the quality of UGS in the survey to comprehend whether changes or enhancements in these factors are conducive to improving women's geographies of fear while using green spaces. All participants gave verbal and written consent to audio-record their interview and focus group session, with the guarantee that their identity would remain anonymous. The ethical standards of this study were approved by the University of Melbourne's Psychology Health and Applied Sciences Human Ethics Sub-Committee (2056618).

## 3. Results

Table 1 presents a summary of the characteristics of the survey respondents.

**Table 1.** Summary of the sociodemographic characteristics of online survey respondents, women living in Mexico City (*n* = 1245) June 2020.

| Age Group | *n* | % |
|---|---|---|
| 18–24 | 412 | 33.1% |
| 25–29 | 154 | 12.4% |
| 30–34 | 109 | 8.8% |
| 35–39 | 88 | 7.1% |
| 40–44 | 84 | 6.7% |
| 45–49 | 76 | 6.1% |
| 50–54 | 94 | 7.6% |
| 55–59 | 86 | 6.9% |
| 60–64 | 77 | 6.2% |
| 65+ | 63 | 5.1% |
| **# of other people living in the house** | | |
| Living alone | 44 | 3.5% |
| Living with 1 person | 242 | 19.4% |
| Living with 2 people | 268 | 21.5% |
| Living with 3 people | 297 | 23.9% |
| Living with 4 people | 180 | 14.5% |
| Living with 5 or more | 214 | 17.2% |

**Table 1.** *Cont.*

| Socio-economic status | | |
|---|---|---|
| Low-income | 507 | 40.7% |
| Middle-income | 284 | 22.8% |
| High-income | 442 | 35.5% |
| **Education** | | |
| High school or less | 201 | 16.1% |
| Technical degree | 128 | 10.3% |
| Undergraduate | 615 | 49.4% |
| Graduate school | 298 | 23.9% |

*3.1. Characteristics of Survey Respondents and UGS Use (Access) during the Pandemic*

Before Mexico City's government introduced restrictions to prevent the spread of COVID-19 on March 21, 2020, 1189 (95.5%) women indicated using UGS. This figure dropped to 700 (56.2%) once restrictions came into place, a decrease of almost 40%. Out of the 700 women who used UGS after restrictions were introduced, the majority (78.4%) chose to use the green space closest to home. Interestingly, 79% of women indicated not having enough UGS in their neighborhood. Of users, almost 10% reported staying at UGS for less than 15 min per visit, 34% between 16 to 30 min per visit, and most users, 56%, indicated using UGS for more than 30 mins per visit on average.

Notably, while only 39% of women living alone reported using UGS during the restrictions, this percentage increased to 57% for women living with someone else. It is also interesting that 62% of low-income women indicated using UGS after restrictions began, compared with 54% of middle-income women and 51% of high-income women. Regarding age, 53% of women aged 65 and older reported having stopped using UGS once the pandemic began, the largest decrease of any group. This result was expected given the vulnerability of this group to complications in the event of contracting COVID. Surprisingly, women aged 30 to 34 years were the second-largest group of lost UGS users, with 46% revealing that they stopped using UGS after restrictions started. In third place were women aged 60 to 64, at 44%. The rest of the age groups suffered a reduction of 36 to 42% in the number of users.

When we asked the 44% of women who specified not using UGS after the pandemic began about their motives, 88% of those women indicated that, due to the health emergency, they preferred to stay at home, either for fear of catching COVID, for fear of infecting someone at home, or, in some cases, because they were ill and needed to quarantine. Safety concerns occupied the second spot, with 26% of respondents who did not use UGS expressing fear of suffering some type of violence while using these spaces. Meanwhile, not having UGS close to home and them being of poor quality was the third (23%) and fourth (14%) most cited reasons. Proximity issues included women's responses that their closest UGS had closed due to pandemic regulations, leaving them with no other nearby options.

When we asked the women who said they had used UGS at some point (either before or after the COVID restrictions began) what activities they carried out within UGS, the activity reported as most common was walking (83%), followed by passive engagement or relaxing (66.2%), playing sports (20.8%), cycling (19.6%), socializing (14.4%), and buying or selling things in the local markets located inside the green spaces (7%). Notably, out of the 1033 women who indicated they liked to walk while visiting UGS, just over one quarter (262) mentioned that they walked with their pets.

*3.2. Association between UGS Use (Access) and Perceived Quality of UGS in the Neighborhood*

Next, we present the logistic regression results of Models 1, 2, and 3 (see Table 2), which show that the perceived quality of UGS in the neighborhood affected UGS use (access)

after COVID restrictions were introduced. The positive association between perceived quality and access to UGS during mobility restriction is slightly reduced—but remains significant—when additional potential confounders were added to the model. In Model 2, after controlling for age group, living arrangement, and income group, the odds of using UGS after the restrictions are about 1.59 times higher among those who reported good quality UGS in their neighborhood relative to those who reported otherwise (95% CI [1.26–2.00]). Moreover, in Model 3, the positive association between perceived quality of UGS in the neighborhood and use of UGS persists after introducing neighborhood factors: the perceived quietness of the neighborhood and perceptions of adequate UGS in the area. The Hosmer–Lemeshow test for Model 2 yielded a $\chi^2$ (8) of 6.09, and for Model 3, a $\chi^2$ (8) of 10.01 and were both insignificant ($p > 0.05$), suggesting that the models fit the data well.

**Table 2.** Odds ratios and 95% confidence intervals for predictor variables associated with UGS use (access) after COVID-19 restrictions were introduced (21 March 2020), online survey of women living in Mexico City ($n = 1245$), June 2020.

| | M1 | | | | M2 | | | | M3 | | | |
|---|---|---|---|---|---|---|---|---|---|---|---|---|
| | OR | $p$ | | 95% CI | OR | $p$ | | 95% CI | OR | $p$ | | 95% CI |
| **Quality of UGS in the neighborhood** | | | | | | | | | | | | |
| Not good | 1 | | | | 1 | | | | 1 | | | |
| Good | 1.62 | 0.00 | *** | (1.30–2.04) | 1.59 | 0.00 | *** | (1.26–2.00) | 1.44 | 0.00 | *** | (1.13–1.83) |
| **Age group** | | | | | | | | | | | | |
| 18–24 | | | | | 1 | | | | 1 | | | |
| 25–34 | | | | | 0.97 | 0.87 | | (0.70–1.34) | 1.00 | 1.00 | | (0.72–1.39) |
| 35–44 | | | | | 1.31 | 0.16 | | (0.90–1.93) | 1.30 | 0.19 | | (0.88–1.91) |
| 45–54 | | | | | 1.23 | 0.30 | | (0.84–1.80) | 1.24 | 0.28 | | (0.84–1.83) |
| 55–64 | | | | | 1.12 | 0.59 | | (0.75–1.66) | 1.07 | 0.75 | | (0.71–1.60) |
| 65+ | | | | | 0.86 | 0.61 | | (0.49–1.53) | 0.78 | 0.40 | | (0.43–1.40) |
| **Living arrangement** | | | | | | | | | | | | |
| Living alone | | | | | 1 | | | | 1 | | | |
| Living with 1 person | | | | | 1.45 | 0.27 | | (0.75–2.82) | 1.51 | 0.23 | | (0.77–2.95) |
| Living with more than 1 person | | | | | 2.04 | 0.03 | ** | (1.08–3.86) | 2.05 | 0.03 | ** | (1.08–3.89) |
| **Income group** | | | | | | | | | | | | |
| Low-income | | | | | 1 | | | | 1 | | | |
| Middle-income | | | | | 0.70 | 0.02 | ** | (0.52–0.96) | 0.67 | 0.01 | *** | (0.49–0.91) |
| High-income | | | | | 0.66 | 0.01 | *** | (0.49–0.88) | 0.61 | 0.00 | *** | (0.45–0.82) |
| **Quiet neighborhood** | | | | | | | | | | | | |
| No | | | | | | | | | 1 | | | |
| Yes | | | | | | | | | 1.42 | 0.04 | ** | (1.03–2.16) |
| **Enough UGS in the neighborhood** | | | | | | | | | | | | |
| Strongly disagree | | | | | | | | | 1 | | | |
| Disagree | | | | | | | | | 1.20 | 0.25 | | (0.88–1.65) |
| Neither agree nor disagree | | | | | | | | | 1.22 | 0.24 | | (0.87–1.72) |
| Agree | | | | | | | | | 1.48 | 0.06 | * | (0.99–2.21) |
| Strongly agree | | | | | | | | | 1.61 | 0.03 | ** | (1.04–2.49) |

Notes: Significance levels *** $p < 0.01$, ** $p < 0.05$, * $p < 0.1$. Model 1: Unadjusted. Model 2: Model 1 adjusted for respondents' age group, living arrangement and income group. Model 3: Model 2 adjusted for neighborhood characteristics. Mc Fadden's R-squared, Model 1:0.010 ($p = 0.000$). Mc Fadden's R-squared, Model 2: 0.025 ($p = 0.000$). Mc Fadden's R-squared, Model 3: 0.035 ($p = 0.000$).

### 3.3. Features Associated with UGS Quality

What are particular features of UGS associated with women reporting having UGS of "good quality" in their neighborhood? Table 3 suggests that cleanliness and maintenance, good lighting, the presence of walls, and having playgrounds or sports facilities are significantly and positively associated with the likelihood of perceiving UGS as having good quality. Interestingly, the presence of police, a standard safety indicator, is not significantly associated with perceived good UGS quality. The Hosmer–Lemeshow test for Model 4 yielded a $\chi^2$ (8) of 6.81 and was insignificant ($p > 0.05$), suggesting that the model fit the data well.

**Table 3.** Model 4: Odds ratios and 95% confidence intervals for predictor variables associated with perceived good quality of UGS in the neighborhood after COVID-19 restrictions were introduced, online survey of women living in Mexico City ($n = 1245$), June 2020.

| Predictors | OR | $p$ | | 95% CI | |
|---|---|---|---|---|---|
| **Clean and well-maintained** | | | | | |
| No | 1 | | | | |
| Yes | 3.79409 | 0.000 | *** | 2.032134 | 7.083748 |
| **Good lighting** | | | | | |
| No | 1 | | | | |
| Yes | 4.386839 | 0.000 | *** | 2.672711 | 7.200311 |
| **Presence of walls** | | | | | |
| No | 1 | | | | |
| Yes | 2.85302 | 0.000 | *** | 2.137672 | 3.80775 |
| **Low noise** | | | | | |
| No | 1 | | | | |
| Yes | 1.316099 | 0.225 | | 0.844746 | 2.050459 |
| **Presence of markets** | | | | | |
| No | 1 | | | | |
| Yes | 0.74423 | 0.106 | | 0.520042 | 1.065064 |
| **Toilets** | | | | | |
| No | 1 | | | | |
| Yes | 1.225118 | 0.173 | | 0.91493 | 1.64047 |
| **Size (ample)** | | | | | |
| No | 1 | | | | |
| Yes | 1.264952 | 0.269 | | 0.833822 | 1.919001 |
| **Presence of events** | | | | | |
| No | 1 | | | | |
| Yes | 0.989428 | 0.948 | | 0.716886 | 1.365583 |
| **Presence of police** | | | | | |
| No | 1 | | | | |
| Yes | 0.86368 | 0.310 | | 0.65088 | 1.146053 |
| **Playgrounds or sports facilities** | | | | | |
| No | 1 | | | | |
| Yes | 5.078593 | 0.000 | *** | 3.824721 | 6.743527 |

McFadden's $R^2$: 0.043 ($p = 0.000$), McKelvey and Zavoina's $R^2$: 0.003 ($p = 0.000$). *** $p < 0.01$.

### 3.4. Qualitative Analysis Results

Through the in-depth interviews carried out in October 2020, we identified three central themes, which were further explored during the November 2021 focus groups to observe changes in participants' behavior. Below are the key reflections from the interviews and focus groups on each central theme and the changes in perceptions once restrictions were eased.

**Theme 1.** *The availability of UGS in the neighborhood was a critical hindrance to women's UGS access (use) due to COVID-19 concerns.*

One of the main barriers affecting women's use of UGS during the pandemic was the lack of available UGS in the neighborhood, either because those neighborhoods currently

lack UGS or due to the restrictions associated with the COVID-19 pandemic, which included the closure of some public spaces. When reviewing our initial results, we noticed that UGS closures were associated with the size of spaces, with several survey respondents stating that small UGS were often closed. However, the interviews highlighted that residents of upper-middle or upper-income neighborhoods, where most green areas are located, were not restricted in their access to UGS. On the contrary, in neighborhoods with high UGS availability, women were more inclined to use these spaces because they remained open, were not saturated, and adherence to physical distancing protocols was easier.

> *"I often used the parks nearby, especially to walk with my dog and clear my head for a while. The parks near our building did not close. Sometimes, police officers make sure that there are not too many people and that those who come to the park wear face masks."* (Participant 10, age 33)

Conversely, in marginalized areas, which already have little UGS availability and where UGS are often small, restrictions on UGS access were introduced to prevent the spread of COVID-19 due to overcrowding concerns. Consequently, it is safe to assume that access was not strictly limited by UGS size but by neighborhoods' UGS availability.

> *"In my neighborhood, their [UGS] use is not allowed at the moment."* (Participant 1, age 20)

Up to October 2020, most interviewees who used UGS limited themselves to visiting the green space closest to their homes, avoiding spending much time outside. The lack of availability of a nearby green space did not appear to be an incentive to use distant spaces. Even when restrictions began to ease, some women decided to walk around the block instead of walking long distances to access UGS in other neighborhoods due to the fear of contagion. These women continued to engage in this practice even though they reported feeling happier when they saw greenery, indicating a greater urgency to increase UGS availability in underserved neighborhoods.

> *"From September* [2020] *onwards, I started going for a walk because I was desperate to avoid being at home all day, but not to the park because it is closed. I usually just walk around the block."* (Participant 4, age 27)

In November 2021, most participants communicated using public spaces more frequently than the year before, including UGS. Nevertheless, focus group discussions revealed that the use of these spaces was still not as frequent as before the pandemic began. In areas with low UGS availability, overcrowding and a lack of adherence to physical distancing profoundly inhibited the use of some UGS for fear of becoming infected or spreading the disease to other household members.

> *"I live with someone who is at higher risk of serious complications if they get sick with COVID-19, and I don't want to expose them."* (Participant 6, age 23)

> *"I still don't go out much. I go out much more than a few months ago, but I am still not entirely comfortable since some people don't wear masks. . . . The park in front [of the house] is sometimes crowded, and there are new variants."* (Participant 2, age 57)

**Theme 2.** *For women, the concept of UGS quality and safety, including street safety, are intertwined.*

Similar to the survey results, the findings from the interviews suggest a strong association between safety and perceived UGS quality. Interviewees indicated that they avoided those spaces in which they did not feel safe (those of low quality) or waited to have company to use them, with "company" referring to male friends or relatives. A year later, the focus groups showed that women's propensity to avoid unsafe or poor quality spaces continued once the restrictions decreased and UGS use had grown.

> *"I cannot say that a place is of good quality if it is not safe. What is more, I am not particularly eager to go if I consider it unsafe."* (Participant 3, age 45)

Interestingly, not only the quality and safety of UGS prevented women from using them, but also the perceived safety of the surrounding streets. In particular, the interviewees emphasized that walking on inadequate roads increased their feelings of vulnerability and discomfort, already exacerbated by the pandemic, which sometimes led them to avoid using UGS. Those participants who described living in low-income neighborhoods, where minimal infrastructure is common, conveyed more intense feelings of discomfort while walking around their respective neighborhoods.

*"The road leading to the park is ugly; I do not feel safe walking there."* (Participant 6, age 23)

In turn, when focus group members discussed what factors could improve the generalized feeling of vulnerability, good lighting was accentuated as a central factor for encouraging UGS use, especially for women who used green areas at dawn or night. It is important to point out that, during the focus groups, a distinction was made that referring to good lighting as an enabler encompassed the green space and the adjacent streets. For example, there are instances where, despite having a green area of perceived good quality nearby, if the streets leading to it are poorly lit, the participant preferred not to use it due to safety concerns.

*"The neighborhood park is not ugly, it's okay, but I hardly use it because I only have time in the mornings, and the street is very dark at that time."* (Participant 7, age 28)

Consistent with the survey results, the interviews showed that maintenance and cleanliness were fundamental to defining good UGS quality. Indeed, if the space was neglected or had a lot of garbage, it was considered a deterrent to its use, predominantly for those women who visited UGS with children. The focus groups stressed that with the removal of some restrictions, and as the spaces became more crowded, the perception of cleanliness and maintenance of various UGS decreased, negatively affecting their use and women's perception of safety.

*"I feel that if the place looks neglected, half abandoned, it is a dangerous area where there may be gangs or drugs are sold since no one is going to monitor or clean the place."* (Participant 12, age 59)

Interviewees also indicated that their perception of UGS quality was associated with the activities that could be carried out in a given space. For example, UGS with running tracks, playgrounds, or basketball courts were generally perceived as high-quality spaces. Through the focus groups, the topic of exercise equipment also emerged. In specific green areas, the city government has installed fitness equipment. The participants mentioned being in favor of the initiative, as it allows them to participate in a wider variety of exercises without having to go to a gym, incentivizing them to be more active. This type of feature is especially appreciated in the case of small UGS that might not be large enough for people to run or engage in other kinds of sports that require larger spaces. The installation of exercise equipment is also linked to feeling safe since it provides a perception of care and maintenance to the area.

*"In the greenway across the street, they put exercise machines, and I use them with my sister and nephews from time to time; they keep us active. It looks nice because they are well cared for and functional."* (Participant 5, age 25)

An issue that was not evaluated in the survey or interviews but that came up in the discussion groups is the presence of homeless people or street vendors in the parks or the streets neighboring UGS. Their appearance seems to be a common hindrance to UGS use, increasing the geographies of fear.

In contrast to findings from earlier studies, such as Navarrete-Hernandez, Vetro, and Concha [74]—where the presence of walls deters UGS use and is negatively related to safety, our survey indicated a positive relationship with quality, which was further explored in the qualitative analysis. The interviews and focus groups established that for women who visit

medium or large UGS, such as Chapultepec Park, especially those accompanied by children or pets, walls provide the feeling of protection against possible road incidents. However, this observation differed for small parks or gardens, where easy access was preferred.

> *"I like that I can see my boys play . . . even if they run, they are in a closed space where they will not be able to run into the street." (Participant 9, age 35)*

**Theme 3.** *Social cohesion increased women's propensity to use UGS in their neighborhoods despite heightened fears of insecurity experienced during the pandemic.*

In general, the interviews conveyed a heightened feeling of insecurity beyond UGS use. The closure of businesses, the low presence of people in the streets, and the uncertainty of the situation increased women's feelings of vulnerability when using any public space. In addition, seven participants expressed that these feelings did not improve from March 2020 to October 2020 due to a widespread perception that crime rates had increased because of the economic instability resulting from the pandemic.

> *"In these months, there have been more assaults. Nothing has happened to me yet, but I watch the news, and the situation looks bad." (Participant 11, age 33)*

Remarkably, eleven out of the twelve women indicated concern about gender violence in the city, with three of them explicitly addressing the number of femicides per day, eleven, a number constantly repeated in the press. News about gender-based violence had increased women's collective anxiety about using public spaces even before the pandemic began, which was expressed in the focus groups. For instance, one of the interviewees mentioned participating in a march to protest gender-based violence, which was attended by thousands of women in March 2020, a few days before the COVID-19 restrictions came into force.

Both focus groups debated the perception of generalized insecurity in the city. In particular, it is important to mention that this perception only seems to affect women when they use public spaces alone—when accompanied by their partner, family, or friends, the anxiety or fear of being attacked decreases considerably. The latter is consistent with the survey results, which showed higher use of UGS among women who live with others. In this sense, participants agreed that public life is lived in a very different way for women, specifying that as a woman, one must live in a state of constant alertness, especially when going out at night, at dawn, or visiting spaces where there are few people. Geographies of fear seem to be a constant in Mexico City, although it is critical to emphasize that the pandemic has intensified those feelings.

> *"I do not go out to walk my dog at night unless my boyfriend is home, and we go together. It is a deal we have, and that way, I feel calmer and enjoy the walk." (Participant 8, age 38)*

Finally, women living in communities with a sense of social cohesion also described using UGS in their neighborhoods more frequently. In this context, participants articulated social cohesion as having a relationship with their neighbors and a strong sense of solidarity between community members. This finding is especially noteworthy since distant spaces, where the participants do not know other users, do not stimulate feelings of trust and, therefore, use. Thus, a sense of community seems crucial to diminish women's geographies of fear. Social cohesion also affects how women perceive the police. On the one hand, women living in neighborhoods with police assigned to that area, who know police officers personally and have a cordial relationship with them, feel more comfortable and safer having officers around. For instance, women living in gated communities recognized police officers as part of their community.

> *"On the block, there are two policemen who make the rounds on their bicycles and also look after people in the park nearby. We know them well; I greet them whenever I am outside watering my plants." (Participant 10, age 33)*

On the other hand, police presence increases stress levels for women who do not see officers as part of the community, as they associate police with corruption, crime, and mistreatment.

> *"It makes me uncomfortable to encounter cops anywhere. A patrol began to guard our neighborhood after COVID-19 to check that everything was fine. It makes me very nervous that cops are around here, especially because sometimes they go into the buildings to use the bathrooms, and I fear they might steal something." (Participant 9, age 35)*

## 4. Discussion

In Latin American cities, the COVID-19 pandemic has exposed systemic inequalities and accentuated pre-existing urban challenges with significant gender dimensions [42,43]. Specifically, in Mexico City, the pandemic has widened gender disparities by increasing women's sense of vulnerability, affecting their opportunities, motivation, and access to UGS. This effect is evidenced by the results of our survey, which show a significant decrease in the use of UGS by women since the pandemic began. Our quantitative and qualitative analyses illustrate that anxiety about contracting or spreading COVID, UGS closures, and increased geographies of fear have been core barriers to women's access to UGS during this period, predominantly for those women living in underserved areas with low availability of UGS and poor street infrastructure. Markedly, the closure of UGS in marginalized neighborhoods during the first phase of the pandemic negatively impacted the mental health of women in these areas by further restricting their exercise and relaxation options, amplifying health disparities [49,84].

Similar to Dunckel Graglia's [75] empirical evidence on the use of public spaces such as the metro in Mexico City, our findings reveal that the fear of violence hinders women's belief in their right to the city and, therefore, their access to UGS. To this end, research on gender inequalities in Latin America suggested that the growing number of femicides in the region, associated with a fear of violence, could be undermining women's desire to leave the house by influencing their perception of safety in public spaces [10,85]. The latter coincides with observations from our interviews and focus groups, where participants expressed a heightened sense of vulnerability linked to a perception of increased criminality. Although the fear of violence when using public spaces is a barrier experienced by women worldwide, gender-based violence is more prevalent in the cities of the Global South, which translates into lower participation rates among women in the urban public sphere [86,87]. For instance, while a recent study of access to public parks and gardens in urban areas of England and Wales revealed that women preferred to visit UGS that were less crowded due to COVID-19 contagion concerns [88], our study uncovered that women in Mexico City, conversely, tend to avoid parks with few people due to the fear of crime. Thus, to increase women's access to UGS both on a day-to-day basis and during a public health emergency, it is necessary to consider expanding not only equity in UGS availability but also equity in social access [69], or what Mazumder [12] has defined as experiential equity.

To achieve experiential equity, it is essential to understand why access to infrastructure is restricted beyond the distribution of public resources [12]. In societies with structural inequities, such as Mexico, the fear of objectification, harassment, assault, and rape restrict women's power to participate in society by excluding them from public spaces or limiting their ability to benefit from urban infrastructure, including environmental amenities [21,73,74]. Having no access to open green spaces to exercise or decompress during the pandemic confinement has been a negative repercussion of gendered geographies in Latin America [75], where women cannot use and enjoy UGS due to safety concerns. Accordingly, exploring pathways to ameliorate the perception of geographies of fear can be a tool for differentiating redistribution in cities and reducing health disparities during catastrophic events [16,76]. If green spaces are not well lit, do not have adequate conditions such as clear pathways, have enclosed areas with blind spots, or seem unwelcoming, they can be perceived as hazardous rather than places that stimulate a healthy urban environment [3,19,89].

In this sense, our results suggest that since a better sense of belonging to one's neighborhood can boost women's participation in the public sphere [90,91], it is imperative to promote programs aimed at improving social cohesion to increase women's confidence in using public spaces and improve their trust in institutions such as the police [19,92,93]. Access to social goods in communities with high levels of social fragmentation tends to greatly disadvantage marginalized groups, such as women [94]. Consequently, strengthening neighborhood belonging and identity, and creating networks that enrich social capital are necessary to guarantee women's right to the city [95].

Our findings also indicate that to expand women's UGS access in Mexico City and, thereby, improve their well-being and enhance their feeling of comfort, it is critical to incorporate quality characteristics to existing UGS, such as exercise equipment or playgrounds [17]. Even more so, to reduce women's feeling of vulnerability, UGS and the surrounding streets must have similar quality elements, such as good lighting, cleanliness, and maintenance. This finding is consistent with the growing body of behavioral research showing that routes to access public spaces are especially important for incorporating women into the public sphere and increasing their opportunities to access health and recreation spaces [69,96]. In the case of cleanliness, maintenance, and good lighting, evidence from other countries, such as Australia and the U.K., highlights these two characteristics as essential to intensify feelings of comfort and safety among women; otherwise, spaces evoke fear of crime [19,62].

It is essential to highlight that the suggestions presented in this study are specific to Mexico City. Although the city shares similar characteristics to other Latin American cities [19], and it is reasonable to argue that some observations might be comparable to women's experiences elsewhere, the results cannot be extrapolated to other settings without additional research. For example, given that the literature on geographies of fear indicates that women in other Latin American cities share a fear of violence when using public spaces [97], policies aimed at improving public lighting might be expected to increase the use of UGS by women in other Latin American cities. However, more evidence is needed to test these hypotheses and assess the extent to which the recommendations of this study are relevant to cities in other Latin American countries. This limitation is related to the study's design, which in its second phase, uses a select number of women to deepen the understanding of women's lack of access to UGS instead of seeking representativeness [98].

Finally, although women's access to UGS during a health crisis can be improved by modifying the existing infrastructure to accommodate their needs [99], access to UGS will not be entirely equitable unless structural inequalities that reinforce gender discrimination are eliminated [72]. Accordingly, studies like this that examine potential design interventions that can affect women's willingness to use UGS in specific contexts, ameliorating their perceived safety and comfort, are necessary to develop evidence-based policies that will increase cities' inclusiveness and environmental justice. However, these efforts alone are not sufficient to eliminate the problem of the geographies of fear [100].

## 5. Conclusions

The present paper investigated women's barriers to UGS access during the COVID-19 pandemic in Latin America, using a mixed-methods approach that included a survey, in-depth interviews, and focus groups. The use of various analysis methods made it possible to incorporate sociocultural perceptions into the data analyses, enriching the robustness of the study findings. Using Mexico City as a case study, our results suggest that, in Latin America, women stopped using UGS or decreased their frequency of use since the COVID restrictions took effect. The three main barriers affecting women's UGS access during the pandemic have been the limited availability of green spaces in some neighborhoods, the poor quality of UGS and surrounding streets, and the lack of social cohesion, which has deeply hurt women's willingness to use UGS during the health emergency.

Notably, the interviews and focus groups conveyed that the quality of UGS and their safety are intrinsically connected to women. In turn, the safety of the route to access UGS

is of the utmost importance to encourage their use, with elements such as good lighting, cleanliness, and maintenance promoting a lesser sense of vulnerability. Moreover, the study results point to a need for better guidelines that enforce physical distance during a disease outbreak or pandemic rather than prohibiting access to UGS altogether, particularly in marginalized areas, as evidence shows that restricting access in underserved neighborhoods only contributes to increasing health disparities.

Studies like this one, which capture the perceptions and motivations of women, are instruments of citizen participation necessary to improve cities' infrastructure and environmental justice. However, the equity of access to UGS cannot be entirely achieved until the structural inequalities that lead to gender violence are eliminated. Ensuring women's social, economic, and political inclusion remains a persistent challenge in cities of Latin America.

**Author Contributions:** Conceptualization, C.M.H. and A.U.; methodology, C.M.H. and A.U.; software, C.M.H.; validation, C.M.H. and A.U.; formal analysis, C.M.H.; investigation, C.M.H.; data curation, C.M.H.; writing—original draft preparation, C.M.H.; writing—review and editing, C.M.H. and A.U.; supervision, A.U.; project administration, C.M.H.; funding acquisition, C.M.H. All authors have read and agreed to the published version of the manuscript.

**Funding:** This research was supported by the School of Geography, Earth, & Atmospheric Sciences at the University of Melbourne.

**Institutional Review Board Statement:** The study was conducted according to the guidelines of the Declaration of Helsinki and approved by the Psychology Health and Applied Sciences Human Ethics Sub-Committee of the University of Melbourne (Ethics ID 2056618, approved 8 May 2020).

**Informed Consent Statement:** Informed consent was obtained from all subjects involved in the study.

**Acknowledgments:** Special thanks to the women who contributed to this study.

**Conflicts of Interest:** The authors declare no conflict of interest.

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
