# Peer review of "Barriers Affecting Women’s Access to Urban Green Spaces during the COVID-19 Pandemic"

_land, doi:10.3390/land11040560_

Round 1
Reviewer 1 Report
Manuscript ID: land-1655060
Title: Effects of the COVID-19 Pandemic on the Use and Perception of Urban Green Space
Authors: Carolina Mayen Huerta, Ariane Utomo
Comments and Suggestions for Authors:
The topic and the idea of the study is very interesting and the main assumptions are valuable, also they corresponds very well to the scope and aims of the journal. However there are some weaknesses in the methodological approach, which reduce the value of the paper and therefore require its significant reorganization in my opinion.
1. The title is short and clear, and in line with the presented study, it explain the studied context.
2. Abstract is generally well constructed, however will need improvement if Authors decide to reorganize the manuscript regarding my suggestions.
Key words are related to the topic.
3. The background of the study presented in the section of Introduction is very deep and important, well organized and divided into subsections, and focuses on many aspects; the description / interpretation is directly related to the proposed scope of the study. Very well presented aspects, also based on many valuable and actual literature items.
4. The aim of the study is not easy to find and recognized in its present form, due to the fact that some parts of it are integrated to the first part of the section of Materials and Methods (lines: 208-216), while research questions which are crucial to the study are listed much earlier in the section of literature review (lines: 66-72).
It must be reorganized and presented in one separate paragraph, clearly stated and defined, after a short justification, and include both research questions 1 and 2. I suggest to create a separate paragraph after the section of literature review (Intriduction) and before the section of Materials and Methods.
5. Materials and Methods – the research design itself, including the proposed methods which are adequate to the proposed scope of research and their character, are well selected, also very well described. Unfortunately, from the methodological point of view, the material based on much differ samples used for two stages of the study – quantitative and qualitative data collection – is not clear and thus insufficient, what is the crucial weakness of the manuscript.
It is generally unclear why Authors compare the data collected from a very cross-sectional study in the first part (total = 1914; completed = 1245) with information gathered from a very small number of respondents in a part of qualitative study (n=12). This approach is inadequate and lowers the value of the paper, especially when the results from the second part of research must/should confirm those from the first part. It includes all Themes presented in subsection 3.2. Qualitative Analysis Results. In my opinion, a comparison and the approach which try to justify the quantitative analysis by qualitative study based on so different samples is impossible and has low scientific value.
At the same time, Authors refer to a similar but only quantitative study conducted by McCormack (n = 12) and Rodriguez-Loureiro et al. (probably it is a paper from 2021 focused on n = 27 – but this literature item is not listed in the section of References!). However, those studies are focused only on a small groups of respondents with no correlations to wider or general samples and data.
6. Regarding the lack of data sufficient for comparison of quantitative and qualitative part of the study, the scope of Discussion is too general, as well as Conclusions.
7. Other comments - the order of data in Table 2 is not clear, the main characteristics of respondents should be presented at first, the characteristics of neighborhoods as the second group of data – in present form both parts are mixed. The order must be improved.
There are unnecessary spaces in the text between the reference numbers.
8. Summing up, the topic and general study design is very interesting and valuable, however the presentation of results, and especially the comparison of results from quantitative and qualitative analysis based on much different research samples has many limitations.
Therefore my suggestion is to focus rather on one stage - e.g. quantitative analysis based on a broad sample and compared with similar studies from other countries, or qualitative analysis based on a small sample and compared with relevant results from other countries - to highlight the value of research from the Mexico City. The manuscript must be reorganize/improve in my opinion, or even divided into two separated papers.
Author Response
We want to extend our gratitude to Reviewer 1 for their generosity in providing their expertise and time to improve our manuscript's quality and clarity. We have addressed the questions and comments as detailed below (our response in red). We have also highlighted the manuscript to indicate where we made these changes.
1.The aim of the study is not easy to find and recognized in its present form, due to the fact that some parts of it are integrated to the first part of the section of Materials and Methods (lines: 208-216), while research questions which are crucial to the study are listed much earlier in the section of literature review (lines: 66-72). It must be reorganized and presented in one separate paragraph, clearly stated and defined, after a short justification, and include both research questions 1 and 2. I suggest creating a separate paragraph after the section of literature review (Introduction) and before the section of Materials and Methods.
Thank you for this recommendation. As suggested, the aim of the study is now presented after the Introduction in the literature review (lines 60-79) in a separate paragraph, which reads:
“Using Mexico City as a case study, this paper presents a women-focused analysis aimed at better understanding the barriers that have affected UGS access in Latin America and, therefore, UGS use during the COVID-19-induced crisis. Two central questions guide our paper: 1) What have been the main barriers to women's UGS access during the pandemic? 2) What factors could help increase women's use of UGS? These two questions are critical to understanding women's perspectives on UGS use and how to improve their experience while using these spaces. Furthermore, they have the potential to inform evidence-based policymaking designed to improve existing green areas and make access to these spaces more equitable in Latin American cities, which is of great relevance during a prolonged public health emergency such as the COVID-19 pandemic. To answer these two questions, we use a sequential mixed-methods approach, encompassing three stages in its data collection process and strategy of analysis, one quantitative and two qualitative (John W. Creswell, Fetters, & Ivankova, 2004). Our research's qualitative components helped further explore the statistical results of the quantitative component, a technique widely used in exploratory studies (John W Creswell & Tashakkori, 2007; O’Cathain, Murphy, & Nicholl, 2010). First, we analyzed quantitative data from an online survey (n=1,245) launched in June 2020. Second, through the analysis of survey results, we developed a qualitative instrument, semi-structured in-depth interviews with twelve women in October 2020. Finally, through the interviews, we were able to identify critical themes that required further exploration, which led us to conduct two focus groups with the same participants in November 2021.”
- Materials and Methods – the research design itself, including the proposed methods which are adequate to the proposed scope of research and their character, are well selected, also very well described. Unfortunately, from the methodological point of view, the material based on much differ samples used for two stages of the study – quantitative and qualitative data collection – is not clear and thus insufficient, what is the crucial weakness of the manuscript.
We used a sequential mixed-methods approach, where data or inferences from data in the first phase of the study are used to develop instruments in the second and third phases QUAN->QUAL->QUAL. This is a common approach, often found in diverse fields of exploratory analyses (Leech, Dellinger, Brannagan, & Tanaka, 2009). Even if samples at the different stages vary, mixed methods add value by increasing the validity of the findings and informing the second data source (J. Creswell, Clark, Gutmann, & Hanson, 2003). Studies using mixed methods often differ largely in sample sizes between their quantitative and qualitative phases. For instance, McKim (2015) used an explanatory mixed-methods analysis to examine the perceived value of mixed methods research for graduate students. While 113 graduate students completed the quantitative component, a survey, only 11 students were selected to participate in two focus groups, the qualitative component of the study.
Similarly, in a mixed-methods study conducted by Ivankova, Creswell, and Stick (2006), 207 participants were included in the quantitative phase of the study, while only four participants were selected for the qualitative phase. Sussman, Williams, Leverence, Gloyd, and Crabtree (2006) carried out a sequential mixed methods design with qualitative assessments (interviews and focus groups). The qualitative analysis included two focus groups conducted with ten clinicians and six interviews. Meanwhile, the quantitative component, a survey, had an N=195. Perry, DeWine, Duffy, and Vance (2007) had similar differences in sample sizes. While the responses of 64 participants were included in the quantitative portion of their study, only four girls and four boys were selected for pre- and postintervention interviews.
It is generally unclear why Authors compare the data collected from a very cross-sectional study in the first part (total = 1914; completed = 1245) with information gathered from a very small number of respondents in a part of qualitative study (n=12). This approach is inadequate and lowers the value of the paper, especially when the results from the second part of research must/should confirm those from the first part. It includes all Themes presented in subsection 3.2. Qualitative Analysis Results. In my opinion, a comparison and the approach which try to justify the quantitative analysis by qualitative study based on so different samples is impossible and has low scientific value.
As mentioned before, we believe that to best understand the issue of women’s lack of access to UGS, combining complementary research methods within a single study can be extremely useful. “Mixed methods research, with its focus on the meaningful integration of both quantitative and qualitative data, can provide a depth and breadth that a single approach may lack by itself” (Burns, 2009). This work focuses on addressing the reasons behind women’s limited UGS use rather than reporting associations. Therefore, the small number of participants in the qualitative phases of the study allowed us to explore in-depth participants’ barriers to UGS use during the pandemic, having prolonged interactions with each participant, delving into observations. Each participant was asked about their behavioral habits during the pandemic (How often do they go out? What places do they frequent the most? What type of activities do they do?). Also, participants were asked about potential anxieties about going out and changes in their interactions with others.
Other studies using a mixed-methods approach also show significant differences in sample sizes. For example, Johnson (2009) used a mixed-methods approach, including 13,000 responses in its quantitative phase and only ten responses from students and five from teachers in its qualitative phase. Guerra, Williams, and Sadek (2011) surveyed 2,678 elementary, middle, and high school youth for the quantitative component of their study, while only interviewing 115 for their qualitative one. In phase 1, Beck’s (2014) quantitative survey collected data from 1,129 respondents. During Phase 2, 12 selected respondents were interviewed. Wallace, Clark, and White (2012) conducted interviews with 18 participants and surveyed 213 people for their mixed-methods study.
Additionally, the number of participants in the qualitative phase is aligned with other studies of this nature:
- 11 participants ages 16 to 27: Gibson, B. E., Mistry, B., Smith, B., Yoshida, K. K., Abbott, D., Lindsay, S., & Hamdani, Y. (2013). The Integrated Use of Audio Diaries, Photography, and Interviews in Research with Disabled Young Men. International Journal of Qualitative Methods, 12(1), 382-402. doi:10.1177/160940691301200118
- 11 participants: Akhtar, S., Dolan, A., & Barlow, J. (2017). Understanding the relationship between state forgiveness and psychological wellbeing: A qualitative study. Journal of religion
- 14 migrant women: Giritli-Nygren, K. and U. Schmauch, picturing inclusive places in segregated spaces: a participatory photo project conducted by migrant women in Sweden. J Gender, Place, Culture, 2012. 19(5): p. 600-614.
- 18 case study households: Thomas, F., Eliciting emotions in HIV/AIDS research: a diary‐based approach. J Area, 2007. 39(1): p. 74-82.
- 30 participants: Reid, Y., Johnson, S., Morant, N., Kuipers, E., Szmukler, G., Thornicroft, G., . . . Prosser, D. (1999). Explanations for stress and satisfaction in mental health professionals: a qualitative study. Soc Psychiatry Psychiatr Epidemiol, 34(6), 301-308. doi:10.1007/s001270050148
At the same time, Authors refer to a similar but only quantitative study conducted by McCormack (n = 12) and Rodriguez-Loureiro et al. (probably it is a paper from 2021 focused on n = 27 – but this literature item is not listed in the section of References!). However, those studies are focused only on a small groups of respondents with no correlations to wider or general samples and data.
Thank you for this observation; the correct citation has now been added: “Comparable studies, such as Noël, Rodriguez-Loureiro, Vanroelen, and Gadeyne (2021) in Belgium and McCormack, Petersen, Naish, Ghoneim, and Doyle-Baker (2022) in Canada, have also used in-depth interviews to better understand the association between UGS use and other variables.”
The study conducted by Noël et al. (2021) incorporated 23 interviews, including four double interviews, with 27 persons lasting on average 17 min (ranging from 7 to 30 min). McCormack et al. (2022) conducted semi-structured interviews to capture the perspectives and perceived experiences of 12 adults in Canada.
Additionally, we reviewed other qualitative studies that examine access to UGS and have a similar number of participants. For instance, a study conducted by Corazon et al. (2019) examines barriers to UGS access for people with disabilities using the responses of 24 participants. Macintyre et al. (2019) investigated experiences in adults (five males and ten females) aged 60 years and over while accessing small urban green spaces in a large UK city. Coventry, Neale, Dyke, Pateman, and Cinderby (2019) used a mixed-methods approach that included 8 participants for the qualitative section of their study, which examines the association between access to public green space and improved mood.
- Regarding the lack of data sufficient for comparison of quantitative and qualitative part of the study, the scope of Discussion is too general, as well as Conclusions.
Thank you so much for this observation. The Discussion now includes a paragraph addressing the study’s limitations (starting in line 669). The paragraph reads:
“It is essential to highlight that the suggestions presented in this study are specific to Mexico City. Although the city shares similar characteristics to other Latin American cities (Sreetheran & van den Bosch, 2014), and it is reasonable to argue that some observations might be comparable to women's experiences elsewhere, the results cannot be extrapolated to other settings without additional research. For example, given that the literature on geographies of fear indicates that women in other Latin American cities share a fear of violence when using public spaces (Dammert, 2012), policies aimed at improving public lighting might be expected to increase the use of UGS by women in other Latin American cities. However, more evidence is needed to test these hypotheses and assess the extent to which the recommendations of this study are relevant to cities in other Latin American countries. This limitation is related to the study's design, which in its second phase uses a select number of women to deepen the understanding of women's lack of access to UGS instead of seeking representativeness (Queirós, Faria, & Almeida, 2017).”
Similarly, the conclusions indicate that these results are explicit to Mexico City:
“Our results revealed that, in Mexico City, women stopped using UGS or decreased their frequency of use since the COVID restrictions took effect. The three main barriers affecting women’s UGS access during the pandemic have been the limited availability of green spaces in some neighborhoods, the poor quality of UGS and surrounding streets, and the lack of social cohesion, which has deeply hurt women’s willingness to use UGS during the health emergency.”
- Other comments - the order of data in Table 2 is not clear, the main characteristics of respondents should be presented at first, the characteristics of neighborhoods as the second group of data – in present form both parts are mixed. The order must be improved.
The models were designed to show the effect of the perceived quality of UGS in the neighborhood (independent variable) on UGS use (access) after COVID restrictions were introduced (dependent variable). The individual (M2) and neighborhood (M3) characteristics are shown as cofounders. The leading association to be measured was between quality and access. We use the convention when presenting a stepwise logistic regression analysis (Wisner, 1990), where the main variable of interest is added in the first model and covariates are added in subsequent models (Pace & Briggs, 2009).
There are unnecessary spaces in the text between the reference numbers. Sorry for this mistake, it has been amended.
- Summing up, the topic and general study design is very interesting and valuable, however the presentation of results, and especially the comparison of results from quantitative and qualitative analysis based on much different research samples have many limitations.
Therefore my suggestion is to focus rather on one stage - e.g. quantitative analysis based on a broad sample and compared with similar studies from other countries, or qualitative analysis based on a small sample and compared with relevant results from other countries - to highlight the value of research from the Mexico City. The manuscript must be reorganize/improve in my opinion, or even divided into two separated papers.
We consider that having a mixed-methods approach is especially beneficial for creating a more holistic picture of a phenomenon by combining the strengths of different research methods (Täuscher & Laudien, 2018). The rationale for including two qualitative components in this study was to deepen the reasons behind the survey’s key findings, enriching the robustness of the study’s insights by ensuring that the follow-up qualitative data provided a better understanding of the survey results (Mao, 2014). Therefore, we would like the manuscript to be considered for publication as a mixed-methods study. We have provided evidence of similar studies in which differences in sample sizes were significant.
References:
Beck, C. D. (2014). Antecedents of Servant Leadership: A Mixed Methods Study. Journal of Leadership & Organizational Studies, 21(3), 299-314. doi:10.1177/1548051814529993
Burns, A. (2009). Mixed Methods. In J. Heigham & R. A. Croker (Eds.), Qualitative Research in Applied Linguistics: A Practical Introduction (pp. 135-161). London: Palgrave Macmillan UK.
Corazon, S. S., Gramkow, M. C., Poulsen, D. V., Lygum, V. L., Zhang, G., & Stigsdotter, U. K. (2019). I Would Really like to Visit the Forest, but it is Just Too Difficult: A Qualitative Study on Mobility Disability and Green Spaces. Scandinavian Journal of Disability Research, 20(1), 1-13. doi:10.16993/sjdr.50
Coventry, P. A., Neale, C., Dyke, A., Pateman, R., & Cinderby, S. (2019). The Mental Health Benefits of Purposeful Activities in Public Green Spaces in Urban and Semi-Urban Neighbourhoods: A Mixed-Methods Pilot and Proof of Concept Study. 16(15), 2712.
Creswell, J., Clark, V. P., Gutmann, M., & Hanson, W. (2003). Handbook of mixed methods in social and behavioral research. Tashakkori A, Teddlie C, editors. In: SAGE Publications.
Creswell, J. W., Fetters, M. D., & Ivankova, N. V. (2004). Designing A Mixed Methods Study In Primary Care. 2(1), 7-12. doi:10.1370/afm.104 %J The Annals of Family Medicine
Creswell, J. W., & Tashakkori, A. (2007). Developing publishable mixed methods manuscripts. In (Vol. 1, pp. 107-111). Journal of Mixed Methods Research: Sage Publications Sage CA: Los Angeles, CA.
Dammert, L. (2012). Fear and Crime in Latin America: Routledge.
Guerra, N. G., Williams, K. R., & Sadek, S. (2011). Understanding bullying and victimization during childhood and adolescence: a mixed methods study. Child Dev, 82(1), 295-310. doi:10.1111/j.1467-8624.2010.01556.x
Ivankova, N. V., Creswell, J. W., & Stick, S. L. (2006). Using Mixed-Methods Sequential Explanatory Design: From Theory to Practice. Field Methods, 18(1), 3-20. doi:10.1177/1525822X05282260
Johnson, L. S. (2009). School contexts and student belonging: A mixed-methods study of an innovative high school. School Community Journal, 19(1), 99-118.
Leech, N. L., Dellinger, A. B., Brannagan, K. B., & Tanaka, H. (2009). Evaluating Mixed Research Studies: A Mixed Methods Approach. Journal of Mixed Methods Research, 4(1), 17-31. doi:10.1177/1558689809345262
Macintyre, V. G., Cotterill, S., Anderson, J., Phillipson, C., Benton, J. S., & French, D. P. (2019). “I Would Never Come Here Because I’ve Got My Own Garden”: Older Adults’ Perceptions of Small Urban Green Spaces. 16(11), 1994.
Mao, J. (2014). Social media for learning: A mixed methods study on high school students’ technology affordances and perspectives. Computers in Human Behavior, 33, 213-223. doi:https://doi.org/10.1016/j.chb.2014.01.002
McCormack, G. R., Petersen, J., Naish, C., Ghoneim, D., & Doyle-Baker, P. K. (2022). Neighbourhood environment facilitators and barriers to outdoor activity during the first wave of the COVID-19 pandemic in Canada: a qualitative study. Cities & Health, 1-13. doi:10.1080/23748834.2021.2016218
McKim, C. A. (2015). The Value of Mixed Methods Research: A Mixed Methods Study. Journal of Mixed Methods Research, 11(2), 202-222. doi:10.1177/1558689815607096
Noël, C., Rodriguez-Loureiro, L., Vanroelen, C., & Gadeyne, S. (2021). Perceived Health Impact and Usage of Public Green Spaces in Brussels' Metropolitan Area During the COVID-19 Epidemic. Frontiers in Sustainable Cities, 3(30). doi:10.3389/frsc.2021.668443
O’Cathain, A., Murphy, E., & Nicholl, J. (2010). Three techniques for integrating data in mixed methods studies. 341, c4587. doi:10.1136/bmj.c4587 %J BMJ
Pace, N. L., & Briggs, W. M. (2009). Stepwise Logistic Regression. Anesthesia & Analgesia, 109(1).
Perry, J. C., DeWine, D. B., Duffy, R. D., & Vance, K. S. (2007). The Academic Self-Efficacy of Urban Youth: A Mixed-Methods Study of a School-to-Work Program. Journal of Career Development, 34(2), 103-126. doi:10.1177/0894845307307470
Queirós, A., Faria, D., & Almeida, F. (2017). Strengths and limitations of qualitative and quantitative research methods. European journal of education studies. doi:http://dx.doi.org/10.46827/ejes.v0i0.1017.
Sreetheran, M., & van den Bosch, C. C. K. (2014). A socio-ecological exploration of fear of crime in urban green spaces – A systematic review. Urban Forestry & Urban Greening, 13(1), 1-18. doi:10.1016/j.ufug.2013.11.006
Sussman, A. L., Williams, R. L., Leverence, R., Gloyd, P. W., Jr., & Crabtree, B. F. (2006). The art and complexity of primary care clinicians' preventive counseling decisions: obesity as a case study. Ann Fam Med, 4(4), 327-333. doi:10.1370/afm.566
Täuscher, K., & Laudien, S. M. (2018). Understanding platform business models: A mixed methods study of marketplaces. European Management Journal, 36(3), 319-329. doi:https://doi.org/10.1016/j.emj.2017.06.005
Wallace, S., Clark, M., & White, J. (2012). ‘It's on my iPhone’: attitudes to the use of mobile computing devices in medical education, a mixed-methods study. 2(4), e001099. doi:10.1136/bmjopen-2012-001099 %J BMJ Open
Wisner, D. H. (1990). A stepwise logistic regression analysis of factors affecting morbidity and mortality after thoracic trauma: effect of epidural analgesia. The Journal of trauma, 30(7), 799-804; discussion 804-795. doi:10.1097/00005373-199007000-00006
Reviewer 2 Report
The paper presents research on the barriers that affect women's access to urban green spaces during the COVID-19 pandemic in Mexico City. It is an interesting paper but I am not convinced how different are the barriers for visiting UGS during pandemic or normality. For instance, poor quality should be always a barrier to not visiting a place. What is the difference during a pandemic? In the end, I am wondering whether these findings would advise differently the policymakers concerning the UGS if the same research have been done for only before pandemic.
it is not clear to me why the authors mention that they use a mixed-method approach for data collection. This would make sense if they had collected i.e. sensors data and narrative data. This is not the case for their methodology so I hesitate to agree with the concept of the mix-method approach.
What is more, I cannot understand why the use of UGS during pandemics could advise stakeholders or urban planners to improve their quality. This shouldn't be the same during pandemic and normality?
How did you use the results of the health-related questions?
Did you normalize the number of responses from the municipalities? The municipality could be a significant confounder so it would make sense to consider the number of responses in relation to the population of each municipality.
How did you select the twelve women for the in-depth interviews> Despite the age criterion? Were there more? What was the procedure? Why did you select twelve?
Lines 368-370: the sentence seems to be incomplete or at least unclear. Please reform.
The authors conclude that the barriers to decreasing or stopping the use of UGS are the limited availability of UGS, their poor quality, and the lack of social cohesion. These barriers shouldn't be the same before the pandemic?
Author Response
We want to extend our gratitude to Reviewer2 for their generosity in providing their expertise and time to improve our manuscript's quality and clarity. We have addressed the questions and comments as detailed below (our response in red). We have also highlighted the manuscript to indicate where we made these changes.
The paper presents research on the barriers that affect women's access to urban green spaces during the COVID-19 pandemic in Mexico City. It is an interesting paper, but I am not convinced how different are the barriers for visiting UGS during pandemic or normality. For instance, poor quality should be always a barrier to not visiting a place. What is the difference during a pandemic? In the end, I am wondering whether these findings would advise differently the policymakers concerning the UGS if the same research have been done for only before pandemic.
In Mexico City and other Latin American cities, the use of UGS has significantly decreased since the pandemic began, suggesting that additional barriers to access UGS were introduced or perceived with this change. As mentioned in the Discussion, the pandemic accentuated pre-existing urban challenges, widening existing inequalities. For instance, although the fear of violence while using UGS was present before the pandemic started, as the participants indicated, UGS tend to be less populated since the introduction of restrictions, making women feel even more at risk while alone. This heightened sense of fear can partially explain the decline of almost 40% in UGS use. In conditions outside the context of the pandemic, social cohesion or street lighting are not absolute impediments to using these spaces; this changed during the pandemic. In addition, the study suggests that the availability of green areas was also altered as several UGS closed, affecting women's access to UGS, mainly those who live in marginalized neighborhoods. These observations are essential for urban planners and policymakers looking to develop resilience strategies for the city in the future.
It is not clear to me why the authors mention that they use a mixed-method approach for data collection. This would make sense if they had collected i.e., sensors data and narrative data. This is not the case for their methodology, so I hesitate to agree with the concept of the mix-method approach.
We used a sequential mixed-methods approach, where data or inferences from the first phase of the study are used to develop instruments in the second and third phases QUAN->QUAL->QUAL. This is a common approach, often found in diverse fields of exploratory analyses (Creswell, Clark, Gutmann, & Hanson, 2003; Leech, Dellinger, Brannagan, & Tanaka, 2009; Perry, DeWine, Duffy, & Vance, 2007). The use of quantitative (survey) and qualitative (in-depth interviews and focus groups) methods to carry out this study qualifies it as a mixed-methods one (Burns, 2009; Ivankova, 2013; Johnson, 2009; Mao, 2014; Täuscher & Laudien, 2018). We consider that having a mixed-methods approach is especially beneficial for creating a more holistic picture of a phenomenon by combining the strengths of different research methods, a survey, focus groups, and interviews (Täuscher & Laudien, 2018). Our rationale for including a qualitative component in this study was to deepen the reasons behind the survey's key findings, enriching the robustness of the study's insights by ensuring that the follow-up qualitative data provided a better understanding of the survey results (Mao, 2014).
What is more, I cannot understand why the use of UGS during pandemics could advise stakeholders or urban planners to improve their quality. This shouldn't be the same during pandemic and normality?
As the paper highlights, before Mexico City's government introduced restrictions to prevent the spread of COVID-19 on March 21, 2020, 1,189 (95.5%) women indicated using UGS. This figure dropped to 700 (56.2%) once restrictions came into place, a decrease of almost 40%. The decline in the use of these spaces contrasts with the increase in UGS use in other cities (Freeman & Eykelbosh, 2020), suggesting that this is an issue that needs to be given priority (Venter, Barton, Gundersen, Figari, & Nowell, 2020). In particular, women's lack of UGS use is particularly worrisome during a pandemic as evidence shows that maintaining access to UGS and encouraging their use are essential measures for amplifying positive health behaviors in urban populations (Douglas, Katikireddi, Taulbut, McKee, & McCartney, 2020).
How did you use the results of the health-related questions?
As indicated in the manuscript, health-related questions were not used in this particular study. We will use these data for a subsequent paper examining the association between physical health and UGS use during the pandemic.
Did you normalize the number of responses from the municipalities? The municipality could be a significant confounder so it would make sense to consider the number of responses in relation to the population of each municipality.
We tested municipalities as co-founders, and they were not significant. This could be because some municipalities are significantly large and contain low-, middle-and high-income neighborhoods, leading to no marked distinction in geographic patterns.
How did you select the twelve women for the in-depth interviews> Despite the age criterion? Were there more? What was the procedure? Why did you select twelve?
Thank you for this observation. We've added additional information to clarify the selection process, starting at line 273:
"We invited survey respondents to participate in in-depth interviews that would take place in October 2020. Twelve women (ages 20 to 59) were selected to participate in the interviews, which lasted between 45 and 80 minutes via telephone. Building on the work of Sargeant (2012), who defines how to ensure the quality of participants in qualitative studies, we selected participants who could best inform our research questions and enhance our understanding of the barriers that affected women's access to UGS during the pandemic. Aligned with grounded theory, we used a maximum variation sampling strategy, selecting women from different social strata living in different city municipalities; some lived alone while others with their families, partners, or friends (Palinkas et al., 2015). Additionally, we asked interested parties if they had young children or lived with older adults to assess whether attitudes changed depending on their care responsibilities. The study aimed to provide a multiplicity of perspectives since the selected women came from different backgrounds and had diverse living conditions."
Other studies using a mixed-methods approach also show similar sample sizes. For example, Johnson (2009) used a mixed-methods approach, including 13,000 responses in its quantitative phase and only ten responses from students and five from teachers in its qualitative phase. Similarly, in a mixed-methods study conducted by Ivankova, Creswell, and Stick (2006), 207 participants were included in the quantitative phase of the study, while only four participants were selected for the qualitative phase. Sussman, Williams, Leverence, Gloyd, and Crabtree (2006) carried out a sequential mixed methods design with qualitative assessments (interviews and focus groups). The qualitative analysis included two focus groups conducted with ten clinicians and six interviews. Meanwhile, the quantitative component, a survey, had an N=195. Perry et al. (2007) had similar differences in sample sizes. While the responses of 64 participants were included in the quantitative portion of their study, only four girls and four boys were selected for pre-and postintervention interviews. In phase 1, Beck's (2014) quantitative survey collected data from 1,129 respondents. During Phase 2, 12 selected respondents were interviewed. Wallace, Clark, and White (2012) conducted interviews with 18 participants and surveyed 213 people for their mixed-methods study.
Moreover, we reviewed other qualitative studies that examine access to UGS and have a similar number of participants. For instance, a study conducted by Corazon et al. (2019) examines barriers to UGS access for people with disabilities using the responses of 24 participants. Macintyre et al. (2019) investigated experiences in adults (five males and ten females) aged 60 years and over while accessing small urban green spaces in a large UK city. Coventry, Neale, Dyke, Pateman, and Cinderby (2019) used a mixed-methods approach that included 8 participants for the qualitative section of their study, which examines the association between access to public green space and improved mood.
Additionally, the number of participants in the qualitative phase is aligned with other studies of this nature:
- 11 participants ages 16 to 27: Gibson, B. E., Mistry, B., Smith, B., Yoshida, K. K., Abbott, D., Lindsay, S., & Hamdani, Y. (2013). The Integrated Use of Audio Diaries, Photography, and Interviews in Research with Disabled Young Men. International Journal of Qualitative Methods, 12(1), 382-402. doi:10.1177/160940691301200118
- 11 participants: Akhtar, S., Dolan, A., & Barlow, J. (2017). Understanding the relationship between state forgiveness and psychological wellbeing: A qualitative study. Journal of religion
- 14 migrant women: Giritli-Nygren, K. and U. Schmauch, picturing inclusive places in segregated spaces: a participatory photo project conducted by migrant women in Sweden. J Gender, Place, Culture, 2012. 19(5): p. 600-614.
- 18 case study households: Thomas, F., Eliciting emotions in HIV/AIDS research: a diary‐based approach. J Area, 2007. 39(1): p. 74-82.
- 30 participants: Reid, Y., Johnson, S., Morant, N., Kuipers, E., Szmukler, G., Thornicroft, G., . . . Prosser, D. (1999). Explanations for stress and satisfaction in mental health professionals: a qualitative study. Soc Psychiatry Psychiatr Epidemiol, 34(6), 301-308. doi:10.1007/s001270050148
Lines 368-370: the sentence seems to be incomplete or at least unclear. Please reform.
Sorry about this mistake. It has been amended, and the sentence now reads: In Model 2, after controlling for age group, living arrangement, and income group, the odds of using UGS after the restrictions are about 1.59 times higher among those who reported good quality UGS in the neighborhood relative to those who reported otherwise (95% CI [1.26-2.00]).
The authors conclude that the barriers to decreasing or stopping the use of UGS are the limited availability of UGS, their poor quality, and the lack of social cohesion. These barriers shouldn't be the same before the pandemic?
The barriers experienced after the pandemic varied, and some took on additional importance given the context of the pandemic. As our survey results show, a significant percentage of women stopped using UGS after March 21, 2020. The decline in women's use of UGS can be partially explained by the variations in perceptions introduced due to the fears and anxieties generated as a result of the pandemic. For example, social cohesion was not an impediment to women's access to UGS before the pandemic. However, once the restrictions came into force, women who lived in neighborhoods with greater social cohesion felt safer and used these spaces more frequently. Instead, the lack of social cohesion became a critical barrier to UGS use after COVID-19.
Beck, C. D. (2014). Antecedents of Servant Leadership: A Mixed Methods Study. Journal of Leadership & Organizational Studies, 21(3), 299-314. doi:10.1177/1548051814529993
Burns, A. (2009). Mixed Methods. In J. Heigham & R. A. Croker (Eds.), Qualitative Research in Applied Linguistics: A Practical Introduction (pp. 135-161). London: Palgrave Macmillan UK.
Corazon, S. S., Gramkow, M. C., Poulsen, D. V., Lygum, V. L., Zhang, G., & Stigsdotter, U. K. (2019). I Would Really like to Visit the Forest, but it is Just Too Difficult: A Qualitative Study on Mobility Disability and Green Spaces. Scandinavian Journal of Disability Research, 20(1), 1-13. doi:10.16993/sjdr.50
Coventry, P. A., Neale, C., Dyke, A., Pateman, R., & Cinderby, S. (2019). The Mental Health Benefits of Purposeful Activities in Public Green Spaces in Urban and Semi-Urban Neighbourhoods: A Mixed-Methods Pilot and Proof of Concept Study. 16(15), 2712.
Creswell, J., Clark, V. P., Gutmann, M., & Hanson, W. (2003). Handbook of mixed methods in social and behavioral research. Tashakkori A, Teddlie C, editors. In: SAGE Publications.
Douglas, M., Katikireddi, S. V., Taulbut, M., McKee, M., & McCartney, G. (2020). Mitigating the wider health effects of covid-19 pandemic response. BMJ, 369, m1557. doi:10.1136/bmj.m1557
Freeman, S., & Eykelbosh, A. (2020). COVID-19 and outdoor safety: Considerations for use of outdoor recreational spaces. National Collaborating Centre for Environmental Health, 829.
Ivankova, N. V. (2013). Implementing Quality Criteria in Designing and Conducting a Sequential QUAN → QUAL Mixed Methods Study of Student Engagement With Learning Applied Research Methods Online. Journal of Mixed Methods Research, 8(1), 25-51. doi:10.1177/1558689813487945
Ivankova, N. V., Creswell, J. W., & Stick, S. L. (2006). Using Mixed-Methods Sequential Explanatory Design: From Theory to Practice. Field Methods, 18(1), 3-20. doi:10.1177/1525822X05282260
Johnson, L. S. (2009). School contexts and student belonging: A mixed methods study of an innovative high school. School Community Journal, 19(1), 99-118.
Leech, N. L., Dellinger, A. B., Brannagan, K. B., & Tanaka, H. (2009). Evaluating Mixed Research Studies: A Mixed Methods Approach. Journal of Mixed Methods Research, 4(1), 17-31. doi:10.1177/1558689809345262
Macintyre, V. G., Cotterill, S., Anderson, J., Phillipson, C., Benton, J. S., & French, D. P. (2019). "I Would Never Come Here Because I've Got My Own Garden": Older Adults' Perceptions of Small Urban Green Spaces. 16(11), 1994.
Mao, J. (2014). Social media for learning: A mixed methods study on high school students' technology affordances and perspectives. Computers in Human Behavior, 33, 213-223. doi:https://doi.org/10.1016/j.chb.2014.01.002
Palinkas, L. A., Horwitz, S. M., Green, C. A., Wisdom, J. P., Duan, N., & Hoagwood, K. (2015). Purposeful Sampling for Qualitative Data Collection and Analysis in Mixed Method Implementation Research. Adm Policy Ment Health, 42(5), 533-544. doi:10.1007/s10488-013-0528-y
Perry, J. C., DeWine, D. B., Duffy, R. D., & Vance, K. S. (2007). The Academic Self-Efficacy of Urban Youth: A Mixed-Methods Study of a School-to-Work Program. Journal of Career Development, 34(2), 103-126. doi:10.1177/0894845307307470
Sargeant, J. (2012). Qualitative research part II: Participants, analysis, and quality assurance. In Journal of graduate medical education (Vol. 4, pp. 1-3): The Accreditation Council for Graduate Medical Education Suite 2000, 515 ….
Sussman, A. L., Williams, R. L., Leverence, R., Gloyd, P. W., Jr., & Crabtree, B. F. (2006). The art and complexity of primary care clinicians' preventive counseling decisions: obesity as a case study. Ann Fam Med, 4(4), 327-333. doi:10.1370/afm.566
Täuscher, K., & Laudien, S. M. (2018). Understanding platform business models: A mixed methods study of marketplaces. European Management Journal, 36(3), 319-329. doi:https://doi.org/10.1016/j.emj.2017.06.005
Venter, Z. S., Barton, D. N., Gundersen, V., Figari, H., & Nowell, M. (2020). Urban nature in a time of crisis: recreational use of green space increases during the COVID-19 outbreak in Oslo, Norway. Environmental Research Letters, 15(10), 104075. doi:10.1088/1748-9326/abb396
Wallace, S., Clark, M., & White, J. (2012). 'It's on my iPhone': attitudes to the use of mobile computing devices in medical education, a mixed-methods study. 2(4), e001099. doi:10.1136/bmjopen-2012-001099 %J BMJ Open
Reviewer 3 Report
This is a very thorough and thoughtful analysis of the barriers that women experience to accessing urban green spaces. The authors have gone beyond the superficial assessment of access defined by distance to green space or the presence of parks in a neighborhood to examine sociological and cultural barriers. The conclusion that having a space is not enough to ensure access is important. Access also requires that people feel safe.
Clarification of the following points would enhance the manuscript:
Line 273 and following. The authors state that the surveys were anonymous, however they reached out to 12 women to invite them to participate in interviews. Please, explain how it was possible to identify participants who were supposedly anonymous.
Line 340 - 349. More information about the structure of the survey, especially what appear to be follow-up questions would make it easier to understand how the data were collected. The authors state "When we asked the 44% of women..." which leaves the reader wondering how the 44% of women were identified and contacted for the follow-up questions.
Author Response
We want to extend our gratitude to Reviewer3 for their generosity in providing their expertise and time to improve our manuscript's quality and clarity. We have addressed the questions and comments as detailed below (our response in red). We have also highlighted the manuscript to indicate where we made these changes.
Clarification of the following points would enhance the manuscript:
Line 273 and following. The authors state that the surveys were anonymous, however, they reached out to 12 women to invite them to participate in interviews. Please, explain how it was possible to identify participants who were supposedly anonymous.
Thank you for this observation. It is important to mention that, as the manuscript states, the participants were not anonymous since we know their names, ages, and sociodemographic characteristics. Nonetheless, the manuscript states that participants were guaranteed that their identity would remain anonymous in the documents used for disseminating the results, which was essential for increasing their level of comfort.
Additionally, we have expanded the section that explains the selection criteria, starting in line 273. The paragraph now makes clear how we reached participants:
“We invited survey respondents to participate in in-depth interviews that would take place in October 2020. Twelve women (ages 20 to 59) were selected to participate in the interviews, which lasted between 45 and 80 minutes via telephone. Building on the work of Sargeant (2012), who defines how to ensure the quality of participants in qualitative studies, we selected participants who could best inform our research questions and enhance our understanding of the barriers that affected women's access to UGS during the pandemic. Aligned with grounded theory, we used a maximum variation sampling strategy, selecting women from different social strata living in different city municipalities; some lived alone while others with their families, partners, or friends (Palinkas et al., 2015). Additionally, we asked interested parties if they had young children or lived with older adults to assess whether attitudes changed depending on their care responsibilities. The study aimed to provide a multiplicity of perspectives since the selected women came from different backgrounds and had diverse living conditions.”
Line 340 - 349. More information about the structure of the survey, especially what appear to be follow-up questions would make it easier to understand how the data were collected. The authors state "When we asked the 44% of women..." which leaves the reader wondering how the 44% of women were identified and contacted for the follow-up questions.
We used a sequential mixed-methods approach, where data or inferences from data in the first phase of the study are used to develop instruments in the second phase. This is a common approach, often found in diverse fields of exploratory analyses (Leech, Dellinger, Brannagan, & Tanaka, 2009). The size of the samples at the different stages varied QUAN (1,245)->QUAL (12)->QUAL (12). This is explained in the manuscript.
Even if samples at the different stages differ, mixed methods add value by increasing the validity of the findings and informing the second data source (Creswell, Clark, Gutmann, & Hanson, 2003). Studies using mixed methods often vary in sample sizes between their quantitative and qualitative phases. For instance, McKim (2015) used an explanatory mixed-methods analysis was to examine the perceived value of mixed methods research for graduate students. While 113 graduate students completed the survey, only 11 students were selected to participate in two focus groups.
Similarly, in a mixed-methods study conducted by Ivankova, Creswell, and Stick (2006), 207 participants were included in the quantitative phase of the study, while only four participants were selected for the qualitative phase. Sussman, Williams, Leverence, Gloyd, and Crabtree (2006) carried out a sequential mixed methods design with qualitative assessments (interviews and focus groups). The qualitative analysis included two focus groups conducted with ten clinicians and six interviews. Meanwhile, the quantitative component, a survey, had an N=195. Perry, DeWine, Duffy, and Vance (2007) had similar differences in sample sizes. While the responses of 64 participants were included in the quantitative portion of their study, only four girls and four boys were selected for pre-and postintervention interviews. Johnson (2009) used a mixed-methods approach, including 13,000 responses in its quantitative phase and only ten responses from students and five from teachers in its qualitative phase. In phase 1, Beck’s (2014) quantitative survey collected data from 1,129 respondents. During Phase 2, 12 selected respondents were interviewed.
References:
Beck, C. D. (2014). Antecedents of Servant Leadership: A Mixed Methods Study. Journal of Leadership & Organizational Studies, 21(3), 299-314. doi:10.1177/1548051814529993
Creswell, J., Clark, V. P., Gutmann, M., & Hanson, W. (2003). Handbook of mixed methods in social and behavioral research. Tashakkori A, Teddlie C, editors. In: SAGE Publications.
Ivankova, N. V., Creswell, J. W., & Stick, S. L. (2006). Using Mixed-Methods Sequential Explanatory Design: From Theory to Practice. Field Methods, 18(1), 3-20. doi:10.1177/1525822X05282260
Johnson, L. S. (2009). School contexts and student belonging: A mixed methods study of an innovative high school. School Community Journal, 19(1), 99-118.
Leech, N. L., Dellinger, A. B., Brannagan, K. B., & Tanaka, H. (2009). Evaluating Mixed Research Studies: A Mixed Methods Approach. Journal of Mixed Methods Research, 4(1), 17-31. doi:10.1177/1558689809345262
McKim, C. A. (2015). The Value of Mixed Methods Research: A Mixed Methods Study. Journal of Mixed Methods Research, 11(2), 202-222. doi:10.1177/1558689815607096
Palinkas, L. A., Horwitz, S. M., Green, C. A., Wisdom, J. P., Duan, N., & Hoagwood, K. (2015). Purposeful Sampling for Qualitative Data Collection and Analysis in Mixed Method Implementation Research. Adm Policy Ment Health, 42(5), 533-544. doi:10.1007/s10488-013-0528-y
Perry, J. C., DeWine, D. B., Duffy, R. D., & Vance, K. S. (2007). The Academic Self-Efficacy of Urban Youth: A Mixed-Methods Study of a School-to-Work Program. Journal of Career Development, 34(2), 103-126. doi:10.1177/0894845307307470
Sargeant, J. (2012). Qualitative research part II: Participants, analysis, and quality assurance. In Journal of graduate medical education (Vol. 4, pp. 1-3): The Accreditation Council for Graduate Medical Education Suite 2000, 515 ….
Sussman, A. L., Williams, R. L., Leverence, R., Gloyd, P. W., Jr., & Crabtree, B. F. (2006). The art and complexity of primary care clinicians' preventive counseling decisions: obesity as a case study. Ann Fam Med, 4(4), 327-333. doi:10.1370/afm.566
Round 2
Reviewer 1 Report
Manuscript ID: land-1655060
Title: Barriers affecting women’s access to urban green spaces during the COVID-19 pandemic
Authors: Carolina Mayen Huerta, Ariane Utomo
I appreciate all corrections made by Authors, they increase the quality of both methodological approach and scientific soundness of studied topic. The improved version present the scope and role of used methods much better:
- the presentation of qualitative analysis is better described and also argued in improved form of manuscript, thus its scope, even if the sample is much lower than in quantitative analysis, can be accepted;
- the obtained results are also better argued in relation to cited literature, what makes the Discussion more completed.
My last suggestions are listed below:
- The subsection describing the main study questions is much better organized. The aim of the study is still a bit hidden in the first sentence (lines 77-79). In my opinion, it should be more directly called by changing the order of main words , e.g. The aim of the study is/was to identify and understand the main barriers that have affected UGS access in Latin America by women in Mexico City during the COVID-19 pandemic.......”, etc. The order of words is crucial to make it very clear for the readers, in my opinion.
- some minor English corrections will help to increase the quality of the manuscript
The manuscript can be published after last minor corrections.
Author Response
Dear Reviewer 1,
We appreciate the time and effort you have dedicated to providing your valuable feedback on our manuscript. We are grateful for the insightful comments on our paper and have incorporated changes to reflect your suggestions (our response in red). The changes are highlighted within the manuscript.
- the presentation of qualitative analysis is better described and also argued in an improved form of the manuscript thus its scope, even if the sample is much lower than in quantitative analysis, can be accepted;
- the obtained results are also better argued in relation to cited literature, which makes the Discussion more complete.
My last suggestions are listed below:
- The subsection describing the main study questions is much better organized. The aim of the study is still a bit hidden in the first sentence (lines 77-79). In my opinion, it should be more directly called by changing the order of main words , e.g. The aim of the study is/was to identify and understand the main barriers that have affected UGS access in Latin America by women in Mexico City during the COVID-19 pandemic.......”, etc. The order of words is crucial to make it very clear for the readers, in my opinion.
Thank you for this suggestion. The text has been changed and now reads: “Using Mexico City as a case study, this study aims to identify and understand the main barriers that have affected women’s UGS access in Latin America and, therefore, UGS use during the COVID-19-induced crisis”
- some minor English corrections will help to increase the quality of the manuscript
Thank you so much for this observation. We have reviewed the manuscript and made the necessary changes to improve its quality.
The manuscript can be published after the last minor corrections.
Reviewer 2 Report
The authors addressed very well my questions and the paper has been significantly improved. I suggest accepting the paper in the current version.
Author Response
Dear Reviewer 2,
We appreciate the time and effort you have dedicated to providing your valuable feedback on our manuscript. We are grateful for the insightful comments on our paper.